# Quantitative single-cell proteomics as a tool to characterize cellular hierarchies

Erwin M. Schoof [1,2,3,4,5,6,7✉], Benjamin Furtwängler [1,2,4,7], Nil Üresin[1,2,4], Nicolas Rapin[1,2,4], Simonas Savickas[3], Coline Gentil[1,2], Eric Lechman [5,6], Ulrich auf dem Keller[3], John E. Dick [5,6] & Bo T. Porse [1,2,4✉]

Large-scale single-cell analyses are of fundamental importance in order to capture biological heterogeneity within complex cell systems, but have largely been limited to RNA-based technologies. Here we present a comprehensive benchmarked experimental and computational workflow, which establishes global single-cell mass spectrometry-based proteomics as a tool for large-scale single-cell analyses. By exploiting a primary leukemia model system, we demonstrate both through pre-enrichment of cell populations and through a non-enriched unbiased approach that our workflow enables the exploration of cellular heterogeneity within this aberrant developmental hierarchy. Our approach is capable of consistently quantifying ~1000 proteins per cell across thousands of individual cells using limited instrument time. Furthermore, we develop a computational workflow (SCeptre) that effectively normalizes the data, integrates available FACS data and facilitates downstream analysis. The approach presented here lays a foundation for implementing global single-cell proteomics studies across the world.

[1] The Finsen Laboratory, Rigshospitalet, Faculty of Health Sciences, University of Copenhagen, Copenhagen, Denmark. [2] Biotech Research and Innovation Centre (BRIC), University of Copenhagen, Copenhagen, Denmark. [3] Department of Biotechnology and Biomedicine, Technical University of Denmark, Lyngby, Denmark. [4] Novo Nordisk Foundation Center for Stem Cell Biology, DanStem, Faculty of Health Sciences, University of Copenhagen, Copenhagen, Denmark. [5] Princess Margaret Cancer Centre, University Health Network, Toronto, Canada. [6] Department of Molecular Genetics, University of Toronto, Toronto, Canada. [7] These authors contributed equally: Erwin M. Schoof, Benjamin Furtwängler. ✉email: erws@dtu.dk; bo.porse@finsenlab.dk

Over the last few years, single-cell molecular approaches such as RNAseq (sc-RNAseq) have revolutionized our understanding of molecular cell biology[1–5]. Single-cell resolution has proven to be of utmost importance, particularly within cancer biology, where it has long been known that tumors consist of a multitude of cell types, all acting in concert[6–9]. Similarly, in mammalian organs such as the hematopoietic system, it is the complex interplay of various cell types and differentiation stages that defines a healthy or malignant state[10–17]. While sc-RNAseq methods have been informative about the RNA landscapes in a plethora of biological systems and have demonstrated high clinical relevance[18–20], their readout is limited as a proxy for protein levels[21,22]. Since proteins are the cellular workhorses, there is much knowledge to be gained from deciphering cellular mechanisms at the protein level, either through enzyme activity, post-translational modifications, or protein degradation/proteolysis. Detection of proteins in single-cells was first enabled by antibody-based technologies like Western blot or flow and mass cytometry; however, these methods depend on the availability of high-quality antibodies and are inherently limited in their multiplexing capacity[23].

Recent advances in liquid chromatography mass spectrometry (LC–MS)-based proteomics methods have mitigated major limitations in the sensitivity and throughput required for LC–MS-based single-cell proteomics (scMS). Budnik and colleagues originally proposed the use of isobaric labeling for single-cell proteomics, called ScoPE-MS[24] and the development was continued with ScoPE2[25]. Their goal was not only to increase throughput of single-cell measurements through multiplexing, but also to make use of a carrier channel to provide more peptide copies (200-cell equivalent), and thus ions for peptide identification in addition to the ions in the low abundant single-cell channels; a similar strategy to other low-input sample measurements in the past[26]. Others have since taken a similar multiplexed approach using a carrier channel in combination with a cutting-edge sample preparation technique known as NanoPOTS, and demonstrated very promising results when sample loss is decreased to a minimum[27–29]. Although label-free approaches have also shown promising results[30–34], their throughput currently lags behind the multiplexed approach. Comprehensive evaluations of multiplexed scMS using an isobaric carrier[29,35,36] have further demonstrated the feasibility of the approach, concluded on the tradeoffs of increasing the level of signal boosting with the isobaric carrier, and indicated the importance of estimating the reliability of protein quantification when implementing the method.

In order for scMS to be a viable alternative to sc-RNAseq, we argue it needs to (1) be able to process thousands of cells in a reasonable timeframe, (2) cover a similar order of magnitude in terms of number of proteins detected, and (3) be easily implementable in a wide range of cellular systems. Consequently, we set out to develop a multiplexed scMS workflow that outperforms existing scMS methods in terms of throughput and proteome depth and can be implemented with commercially available resources. To determine whether our experimental workflow would be able to detect biologically relevant cellular heterogeneity within a complex cell mixture, we use a primary Acute Myeloid Leukemia (AML) culture model, termed OCI-AML8227[37] (Fig. 1a). This model maintains the hierarchical nature of AML where a small population of self-renewing leukemic stem cells (LSC; CD34 + CD38−) differentiate to progenitors (CD34 + CD38 + ), that are unable to sustain long-term self-renewal, and finally to terminally differentiated blasts (CD34-). The OCI-AML8227 model system provides us with an ideal proof-of-concept system, as the inherent functional heterogeneity across differentiation stages has previously been evaluated and is readily isolated through FACS sorting based on classical CD34/ CD38 stem cell markers[37–39]. Recapitulating these functional differences using our molecular data would provide proof-of-principle that our workflow is able to distinguish differentiation stages in a complex cellular hierarchy.

Here, we show an experimental workflow that allows global characterization of single-cell proteomes without relying on antibodies for protein identification, and conducted a proof-of-concept study in a primary AML hierarchy. Since multiplexed scMS data present challenges for computational data analysis and should ideally be processed in a streamlined and reproducible manner, we develop SCeptre (Single Cell proteomics readout of expression); a python package tightly integrated with Scanpy[40], that enables quality control, normalization of batch effects and biological interrogation of multiplexed scMS data. The method presented here is inspired by the initial ScoPE-MS efforts and has evolved to cater to: (1) higher throughput characterizations, (2) maximum quantitative accuracy, (3) integrating FACS data from single-cell sorts, and (4) providing a computational workflow for analysis of resulting scMS data and for deciphering cellular heterogeneity.

## Results

**Experimental workflow.** Given the ease by which its distinct subpopulations can be isolated, we reasoned that the OCI-AML8227 model system was ideal for the development and showcasing of an easy-implementable scMS approach (Fig. 1b). Our standard workflow consists of a series of steps. First, single-cells were FACS sorted into individual wells of a 384-well PCR plate containing lysis buffer (Fig. 1 c, d). An important feature in our workflow is the recording of the FACS parameters of each individual cell (termed index-sorting) and the integration thereof during data analysis. Furthermore, a key difference to ScoPE2 is that we use a Trifluoroethanol (TFE)-based lysis buffer, including reduction and alkylation reagents, rather than pure water. Given the chaotropic nature of this reagent[41,42], cell lysis should be more efficient and, in our hands, produced more protein and especially peptide identifications than pure water (Supplementary Fig. 1). Next, cells were lysed through in-plate freezing and boiling, and following overnight digestion, single-cells were labeled using the 16-plex TMTPro technology[43,44]. The 127 C channel is left empty due to isotopic impurity contaminations from the 126 booster channel[25,35] (Supplementary Fig. 2). In our initial experiments, we distributed the remaining fourteen available TMT channels across the three differentiation stages, resulting in five LSC, five progenitors, and four blasts per sample. The booster was prepared separately by sorting 500 cells into each well of a dedicated 384-well plate, followed by the same preparation steps as for the single-cell plates. The individual wells of the booster plate were then pooled in a cell-specific manner to create booster aliquots for the respective cell types. In our initial experiments, we subsequently opted to make a 1:1:1 equimolar booster mix of blast, progenitor, and LSC cells to ensure a homogenous peptide mixture representative of all cell differentiation stages included in our study (see Fig. 1c and Supplementary Data 1 for exact sorting layout). To eliminate the need to clean-up the single-cell samples, we got rid of any non-volatile salts from the buffers. However, initial tests without C18 clean-up of the booster revealed frequent clogging of the analytical LC column, most likely due to cellular debris. Therefore, the booster aliquot was cleaned up using C18-based Stagetip technology[45]. In the following step, the 14 single-cells were pooled and combined with a 200-cell equivalent from the booster aliquot; this level of boosting was previously determined to strike a good balance between proteome depth and quantitative performance[36]. Finally, the sample was dried down using vacuum centrifugation prior to

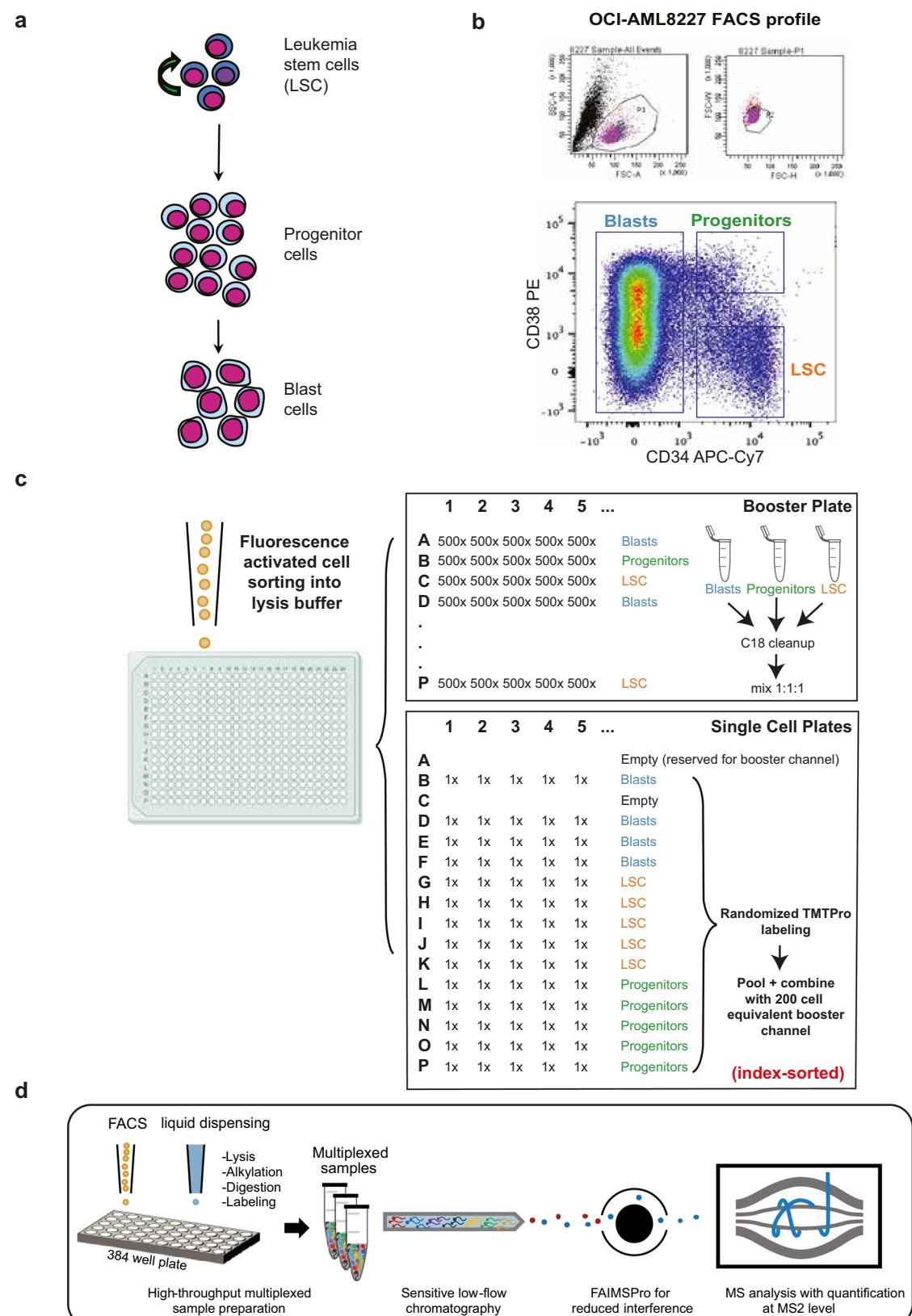

**Fig. 1 Experimental overview of our scMS workflow. a** Overview of the hierarchical nature of an Acute Myeloid Leukemia hierarchy, with leukemic stem cells (LSC) at the apex, differentiating into progenitors, and subsequently, blasts. **b** FACS plot of the OCI-AML8227 hierarchy according to their CD34/CD38 surface marker expression levels. P1 are cells deemed live, P2 excludes doublets and Blasts, Progenitors and LSC are annotated according to CD34/CD38 expression. **c** scMS sample creation overview of booster channel samples and single cells; single-cell TMTpro samples were created with four Blast, five LSC and five Progenitor cells in each pool, labeled randomly using fourteen available TMTpro channels before pooling with a 200-cell equivalent of the 126-labeled booster sample. **d** Conceptual overview of our scMS experimental pipeline; single cells are sorted into 384-well plates containing 1ul of lysis buffer, then digested, TMT labeled and multiplexed. Resulting samples are analyzed with LC–MS via FAIMSPro gas-phase fractionation and Orbitrap detection.

LC–MS analysis. Every 384-well plate thus gives rise to 24 samples, representing the analysis of 336 single-cells per plate. For LC–MS analysis, we used a standard EASY-Spray trap column LC-setup with relatively low-flow (100 nl/min) and a 3 h LC method, coupled to an Orbitrap Exploris™ 480 MS with gas-phase fractionation provided by the FAIMS Pro instrument interface. This device not only filters out contaminating +1 ion species (i.e non-peptide contaminants), but it also switches, on-the-fly, between multiple compensation voltages (CV), each isolating different ion (i.e. peptide) populations, and has thereby shown to lead to greater peptidome and proteome depth, and in addition, to lower levels of co-isolation interference, through the decreased complexity of each gas-phase fraction[46,47]. This LC–MS setup enables a throughput of 112 cells per day, given that 14 cells are analyzed per sample.

**Evaluating the quantitative performance of a booster-based scMS workflow**. In order to strike a favorable balance between proteome depth and quantitative performance (i.e. accuracy and precision), we next investigated appropriate MS instrument settings. With the extremely low peptide amounts from single cells, it is imperative to reach sufficient signal-to-noise (s/n) in the single-cell channels to ensure accurate quantification. On Orbitrap-based instruments, this is commonly achieved by using long injection times (IT) and high corresponding automated gain control (AGC) target values, which results in the collection of large ion populations and robust ion counting statistics[35]. To investigate this aspect in more depth, we generated 12 single-cell samples (each consisting of 14 single cells across all three differentiation stages plus a 200-cell booster) and pooled them into one aliquot, thereby creating 12 technical replicates for LC-MS analysis.

The technical replicate aliquot was injected in triplicate, using IT settings of either 150 ms (150% AGC), 300 ms (300% AGC), 500 ms (500% AGC), or 1000 ms (500% AGC), in order to evaluate the impact of increased IT and AGC target values on quantitative performance. Depicted as a cartoon in Fig. 2a, higher IT/AGC targets, in principle, samples a larger portion of the available ion pool, and thus more closely resembles the true signal. However, this comes with a cost in terms of scan speed due to the longer cycle time, and is therefore expected to result in lower proteome depth. Nevertheless, it should lead to improved s/n values, and consequently, improved quantitative accuracy, precision and sensitivity. In order to interrogate this more closely, we investigated the s/n values on protein level and coefficients of variation (CV) thereof between the triplicates injections across the range of instrument settings used. As shown in Fig. 2b, a clear improvement in overall s/n values is observed when greater IT/AGC target values are used, which conversely reduces the CV between measurements (i.e. improved precision) (Fig. 2b, c). Table 1 shows that for all settings except of the 1000 ms setting, AGC target was not reached in >98% of MS2 scans, and thus the ion collection was only limited by absolute IT rather than ion count. Importantly, the percentage of proteins displaying a CV < 20% was 14%, 30%, 41%, and 49% for 150 ms, 300 ms, 500 ms, and 1000 ms respectively, indicating the clear advantage of using longer IT. However, the trade-off with the sequencing speed becomes apparent for IT higher than 300 ms as the overall lower number of acquired spectra, at a constant peptide-spectrum match (PSM) rate, significantly reduced the absolute number of proteins and the number of proteins with CV below 20%. To evaluate the quantitative accuracy, we calculated protein fold changes between blasts and LSCs in the scMS data using the mean of the populations, and compared them to bulk-sorted OCI-AML8227 MS3-level quantification data. For the latter

dataset, LSCs, progenitors, and blasts were sorted in bulk (20,000 cells per population, in triplicate), labeled with TMTPro as 9-plex, and subjected to high pH fractionation prior to analysis on an Orbitrap Fusion, operating in SPS MS3 mode[48]. This resulted in a bulk proteomics dataset of 6,851 proteins, and provided us with a reference set to compare our single-cell data to in terms of accuracy. This comparison showed that the longer ITs result in a higher correlation of fold changes in scMS with bulk measurements, and further highlighted that when comparing the same high-coverage proteins, correlations did not increase beyond 300 ms, indicating a saturation effect for these presumably high abundant proteins (Fig. 2d, Supplementary Fig. 3). Finally, we evaluated the overall capability of the scMS data to separate blasts from LSCs by calculating the silhouette coefficients of these cells in principle component (PC) space (Methods, Fig. 2e) which, similar to the accuracy, improved with increased ITs and seemed to reach a plateau for the high-coverage proteins. Taken together, these results indicate that both 300 ms and 500 ms could result in biologically meaningful data, and that both settings strike a good balance between proteome depth and quantitative performance when applied to real scMS samples.

**SCeptre, a computational workflow for the analysis of scMS data**. To evaluate both the interrogative capacity of our experimental workflow and the impact of the 'medium' (300 ms) and 'high' (500 ms) instrument parameters, we analyzed 24 samples (one 384-well plate) for each method (Table 2, Supplementary Table 1). To be able to comprehensively analyze the resulting data, we developed SCeptre (Single Cell proteomics readout of expression), a python package that extends the functionalities of Scanpy to process scMS data. As input, SCeptre takes result files from Proteome Discoverer and meta information of the individual cells, a key feature to link recorded FACS parameters back to each cell. First, the individual LC–MS measurements were subjected to quality control to ensure that MS performance remained constant during data acquisition. Next, data was normalized and batch corrected. With TMT data compiled over multiple samples, batch effects can occur on a global level as small differences in sample loading and LC–MS performance shift the overall measured intensities. Furthermore, differential peak sampling and selection of different peptides for protein quantification between sample injections can also lead to protein specific batch effects. A reference or bridge channel is commonly used to normalize the intensities across sample injections on individual protein level[49,50], and has been proposed for scMS analysis[25]. We however decided to implement a normalization strategy that does not rely on a reference channel, since we in our initial experiments employed an actively balanced sample layout, where each sample contains single cells from all the differentiation stages to be interrogated. This ensures the true median protein intensity in each sample to be constant across measurements and enabled protein-specific batch effects between samples to be corrected by equalizing the medians (i.e. one correction factor per protein per sample). Another source of batch effects originates from the different TMT channels, which can be subject to technical bias. For example, the 127 N & 128 C channel can still be affected by 126 booster impurities, resulting in a shift of protein intensities specific to the composition of the booster (Supplementary Fig. 2). Since we randomized the labeling of the cell differentiation stages across channels in all samples, we can also apply the same normalization strategy across channels (i.e. one correction factor per protein per channel). This simple normalization strategy effectively removed batch effects while retaining the biological variability, as it does not rely on the definition of biological covariates (Supplementary Fig. 4a). Previously, a decrease in quantitative

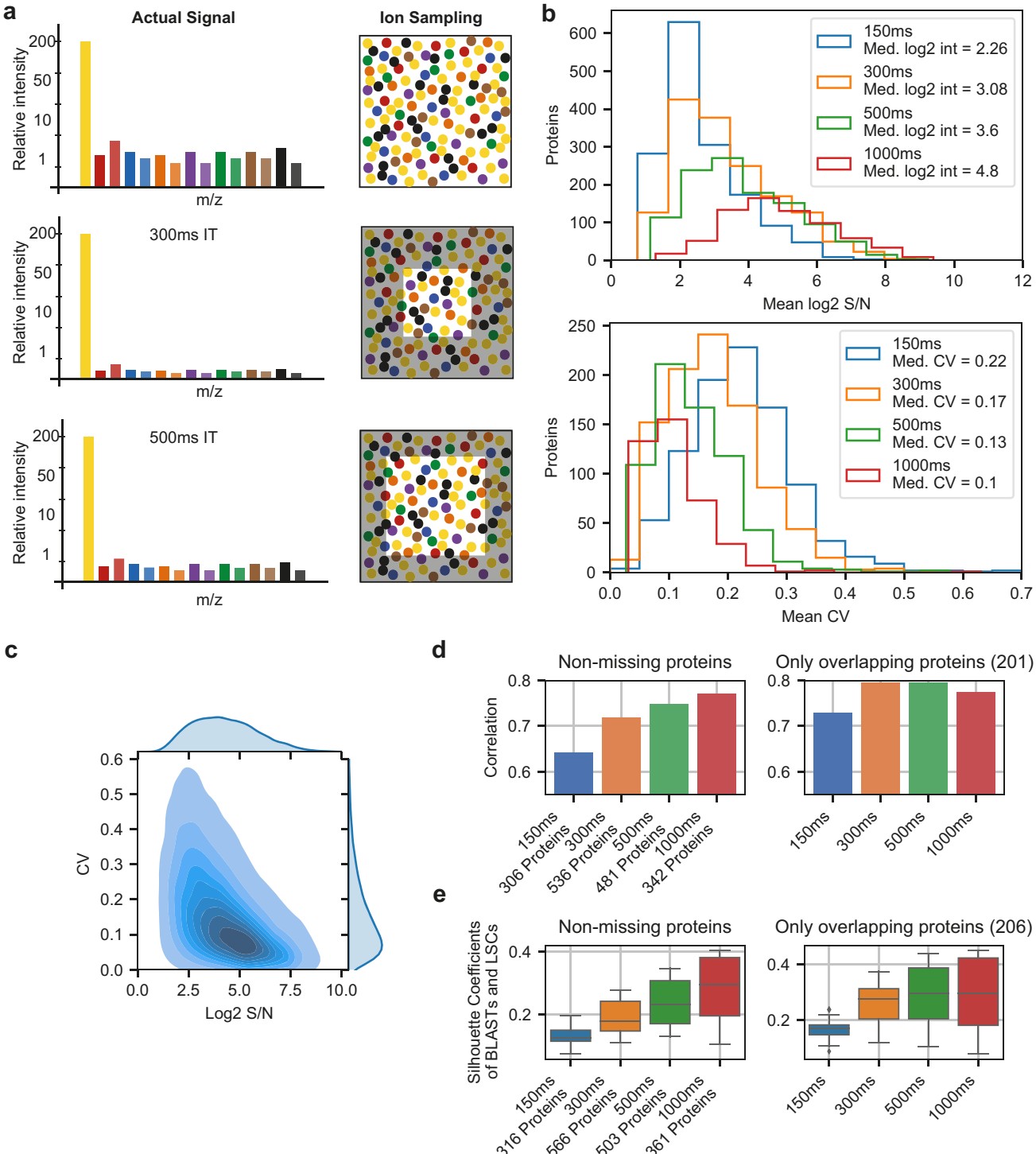

**Fig. 2 Evaluating the quantitative accuracy of a booster-based scMS workflow. a** Cartoon depicting the influence of ion sampling (injection time, IT) on single-cell signal. **b** Evaluation of the effect of increased ion sampling. Top: Histograms of the mean log2 signal-to-noise (s/n) values of all single-cell channel measurements per protein for all four methods. Bottom: Histograms of the mean Coefficient of Variation (CV) per protein for all four methods. Up to 14 CVs per protein were calculated by normalizing the three replicates of each method by equalizing the median s/n of the single-cell channels for each protein across replicates and dividing the standard deviation of the normalized s/n of each protein in each single-cell channel across replicates by the mean of the raw s/n for each protein in each single-cell channel across replicates. The mean CV of up to 14 CVs per protein was reported, as CV calculation was only performed for $n = 3$. **c** Density plot of protein log2 s/n values and their CV from all four methods (i.e. up to 14 CVs per protein per method). **d** Pearson correlation coefficient of fold changes between LSC and blast in single-cell samples and MS3-level bulk data ($n$=number of proteins). Left: For each method, only proteins without missing values were considered (i.e. proteins quantified in all 14 channels in each replicate). Right: Only proteins overlapping between all methods in left were considered. Fold changes in single-cell samples were calculated from the means of all LSCs ($n = 15$) and blast ($n = 12$). **e** Silhouette coefficients of LSC ($n = 15$) and blast ($n = 12$) for each method, calculated in PCA space using the same protein selection as in panel d. Boxplot shows median, 0.25 and 0.75 quantile, and whiskers extend to points within 1.5 interquartile range of lower and upper quartile. Source data are provided as a Source Data file.

**Table 1 Comparison of different IT settings using technical replicates.**

| | MS/MS Spectra | PSM rate (%) | PSMs w. AGC target reached (%) | Protein IDs | Protein IDs w. mean CV < 20% |
|---|---|---|---|---|---|
| 150 ms | 33,002 | 14 | 0.5 | 1538 | 212 |
| 300 ms | 22,418 | 22 | 1.3 | 1546 | 462 |
| 500 ms | 14,556 | 24 | 1.2 | 1110 | 459 |
| 1000 ms | 8,175 | 24 | 14.2 | 692 | 341 |

**Table 2 Comparison of 'medium' and 'high' method using single-cell samples.**

| | Medium (300 ms) | High (500 ms) |
|---|---|---|
| Protein IDs | 2,076 | 1,790 |
| Peptide IDs | 11,111 | 8,826 |
| PSMs | 127,196 | 89,939 |
| PSM rate (%) | 24 | 25.4 |
| Median s/n in single-cell channels | 5.4 | 8.75 |
| Mean protein IDs per file | 1,287 | 999 |
| Filtered cells | 302 | 255 |
| Filtered proteins | 1,986 | 1,498 |
| Mean protein IDs per cell | 1,076 | 712.3 |
| Missing values (%) | 45.8 | 52.45 |

performance of the channels adjacent to the booster channels was observed[35]. However, we decided to still utilize these channels as no batch effects were observed post-normalization. Following the normalization step of SCeptre, individual cells were subjected to quality control. Specifically, we found that the total summed intensity per cell is a suitable parameter to detect outlier cells that could originate from duplets, empty wells, or sample loss during preparation (Supplementary Fig. 4b). After normalization and filtering, the removal of systematic technical biases was verified by evaluating cell-specific parameters as summed intensity and number of genes across technical parameters as sample, channel and row number on the 384-well plate. The importance of this rigorous quality control was exemplified when we observed a row-wise batch effect from the sample preparation that resulted in a separate cluster of progenitors, which could not be corrected as the row number on the 384-well plate confounded with the biological variable of cell differentiation stage (Supplementary Fig. 4c). Removing such cells is crucial to avoid misleading biological conclusions. Subsequently, a median shift of total intensity across cells was applied to normalize for cell size and sampling depth, and finally, data were log2 transformed, yielding the final protein expression matrix. In order to embed the high-dimensional data into fewer dimensions, missing values have to be imputed. However, imputation can introduce noise, especially from low-coverage proteins, as they tend to be of lower abundance, given that total intensity (i.e. s/n) is used as abundance-proxy (Supplementary Fig. 5a). We decided on a minimal protein coverage threshold to exclude low-coverage proteins from contributing to the embedding. This threshold was determined for each dataset using the resulting silhouette score (mean of all silhouette coefficients) in UMAP space after imputation (Methods, Supplementary Fig. 5b). We note that this thresholding should be carefully evaluated on a case-by-case basis, which is facilitated by the SCeptre package. Finally, missing values of the

remaining proteins were imputed using the nearest neighbor algorithm and subsequently scaled to unit variance and zero mean. Collectively, the SCeptre computational workflow provides a framework for the analysis of scMS data.

**scMS enables detection of cellular heterogeneity in an AML hierarchy.** We next wanted to test the ability of our experimental and computational workflows to detect known heterogeneity in the OCI-AML8227 model system. To that end, we compared the 'high' and 'medium' dataset, and found that both methods resulted in good separation of the differentiation stages (Fig. 3a). However, the 'high' method outperformed the 'medium' method in terms of separation power (silhouette coefficients) (Fig. 3b). To avoid bias introduced by different numbers of proteins and cells between the datasets, we repeated the analysis using the subset of overlapping proteins and equalized the cell number per cell type, which produced similar results (Supplementary Fig. 5d, e). Furthermore, the 'high' dataset showed higher correlations of fold changes to bulk data, both across all proteins (Fig. 3c) and for the most abundant proteins (Fig. 3d). We found that the accuracy of the fold changes decreased for proteins with lower s/n (Fig. 3e), however with the 'high' method outperforming the 'medium' method across the entire intensity range. Overall, fold changes measured by scMS tend to be lower compared to bulk measurements, indicating a limited dynamic range (Fig. 3f). However, the 'high' dataset seems to suffer less from these issues. Taken together, these results show that our workflow is able to effectively process multiplexed scMS data, which enabled separation of the known differentiation stages present in the OCI-AML8227 model system.

**Extracting biological information from scMS data.** The apparent improved quantitative accuracy of the 'high' method convinced us to focus on this dataset for the initial biological interrogation of the scMS data. Overlaying the FACS derived intensities of the CD34 and CD38 surface markers on each cell demonstrated that the scMS data can reproduce the observations of the FACS analysis by separating the three differentiation stages while also showing a slight separation of CD38 + and CD38-blasts (Fig. 4a, Supplementary Fig. 6). The successful separation of cell differentiation stages, combined with a near-perfect recapitulation of the cell distribution as defined by the FACS analysis (i.e., CD34/CD38 fluorophore levels) encouraged us to investigate differential protein expression across groups of cells. Here, we made use of the normalized protein expression matrix, while ignoring missing values. We compared blasts against the rest of the cells, which resulted in the identification of differentially expressed (DE) proteins for blasts and the combined group of LSC & progenitors (Fig. 4b, Supplementary Data 2). Gene term enrichment analysis of differentially expressed proteins revealed enrichment of terms associated with protein translation in LSC & progenitors whereas terms associated with myeloid differentiation were enriched in blasts (Supplementary Data 2). Interestingly, we found that some of the DE proteins were expressed heterogeneously in the respective cell differentiation stages, indicating that our scMS data captures gradual protein expression changes that would likely not be detected in bulk data (Fig. 4c). Taken together, these results show that separation of the differentiation stages is driven by biologically meaningful protein-level signal and that our workflow is able to uncover some of these changing proteins.

**Unbiased discovery-based multiplexed single-cell proteomics.** Motivated by these results, we next sought to further characterize the OCI-AML8227 model and test our methods using

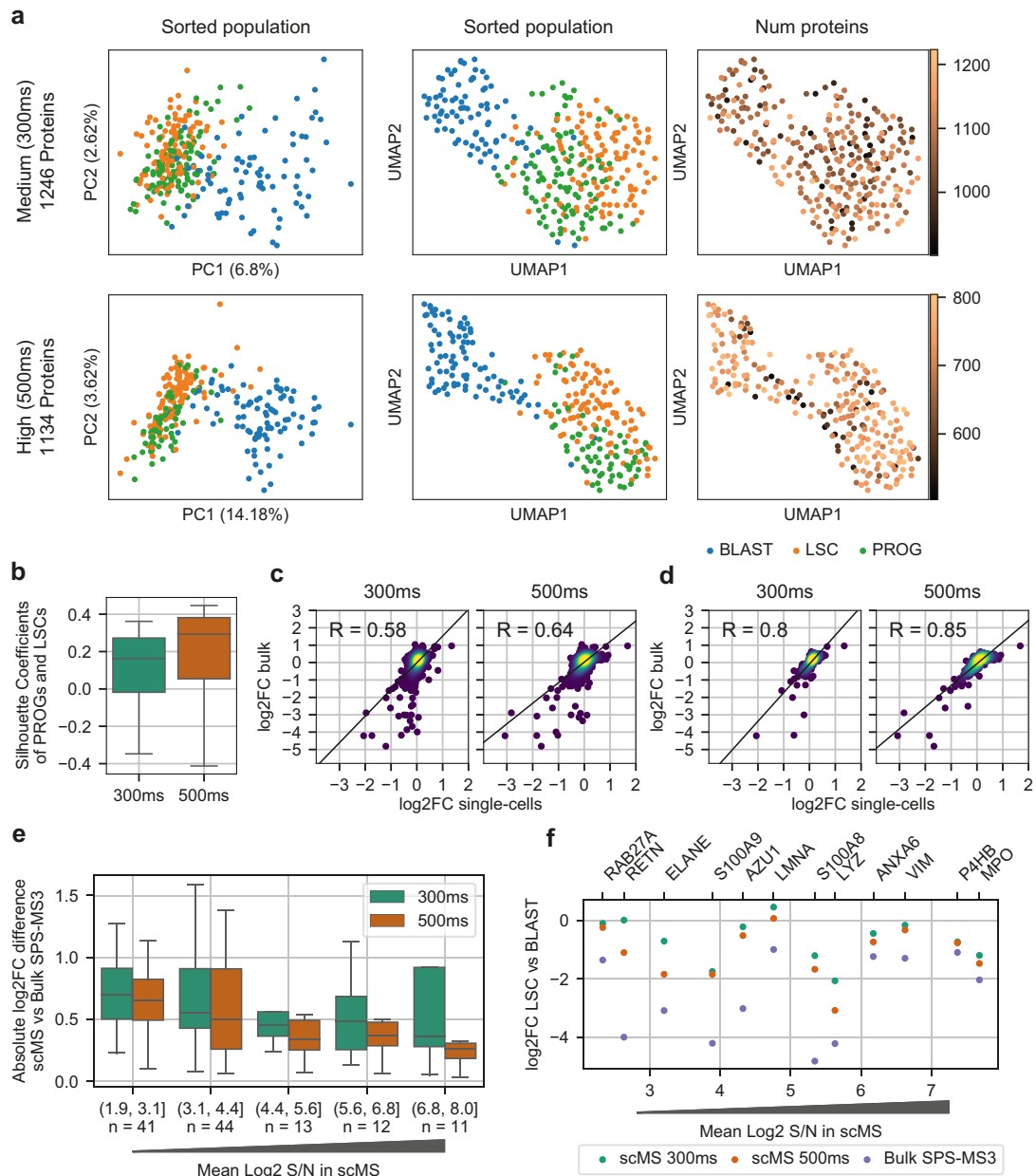

**Fig. 3 Detection of cellular heterogeneity with the 'medium' and 'high' method. a** Comparison of the PCA and UMAP embeddings resulting from the imputed protein matrix. Number of proteins refers to the non-imputed proteins measured per cell. 'Medium'=302 cells, 'high'=255 cells. BLAST = blasts, LSC = leukemia stem cells, PROG = progenitors. **b** Separation of populations measured by the silhouette coefficients of progenitors and LSCs calculated in UMAP space for 300 ms (*n* = 219) and 500 ms (*n* = 166). Boxplot shows median, 0.25 and 0.75 quantile, and whiskers extend to points within 1.5 interquartile range of lower and upper quartile. **c** Pearson correlation of the protein fold changes (log2FC) between blasts and LSCs measured in the scMS workflow and MS3 bulk-sorted data. Non-imputed values were used and only proteins with ≥3 values in blasts and LSCs respectively were considered. Number of proteins in 300 ms and 500 ms is *n* = 1725 and *n* = 1342, respectively. **d** Same analysis as in panel c, but with the top 400 high-coverage proteins selected for each dataset. **e** Absolute log2FC difference of proteins of LSC vs. blast between scMS and MS3 bulk-sorted data across intensity bins. Only proteins that were significantly changed between LSC and blast in the MS3 bulk-sorted data were selected for comparison (FDR < 0.05, absolute log2FC > 0.5). Proteins were binned across their mean log2 signal-to-noise (S/N) in the 300 ms and 500 ms dataset. n= number of proteins with absolute fold change difference plotted. Boxplot shows median, 0.25 and 0.75 quantile, and whiskers extend to points within 1.5 interquartile range of lower and upper quartile. Outliers not shown. **f** log2FC of selected proteins in 300 ms, 500 ms, and MS3 bulk-sorted data. Proteins were binned into 12 bins across their mean log2 S/N in the 300 ms and 500 ms dataset. From each bin the protein with the highest absolute fold-change between LSC and blast in the MS3 bulk-sorted data was selected for comparison with scMS ratios. Source data are provided as a Source Data file.

multiplexed scMS in an unbiased manner by sorting single cells in bulk, without the preselection of specific populations. Additionally, we tested the use of a bulk booster, as the 1:1:1 booster requires prior knowledge about the different populations, which is impractical for discovery-based experiments. Moreover, we introduced empty wells into parts of the dataset to investigate our ability to detect them in SCeptre. Lastly, we used the 127 N channel for a reference channel containing a 10-cell equivalent with the same composition as the booster, to investigate the need for and utility of using it for normalization.

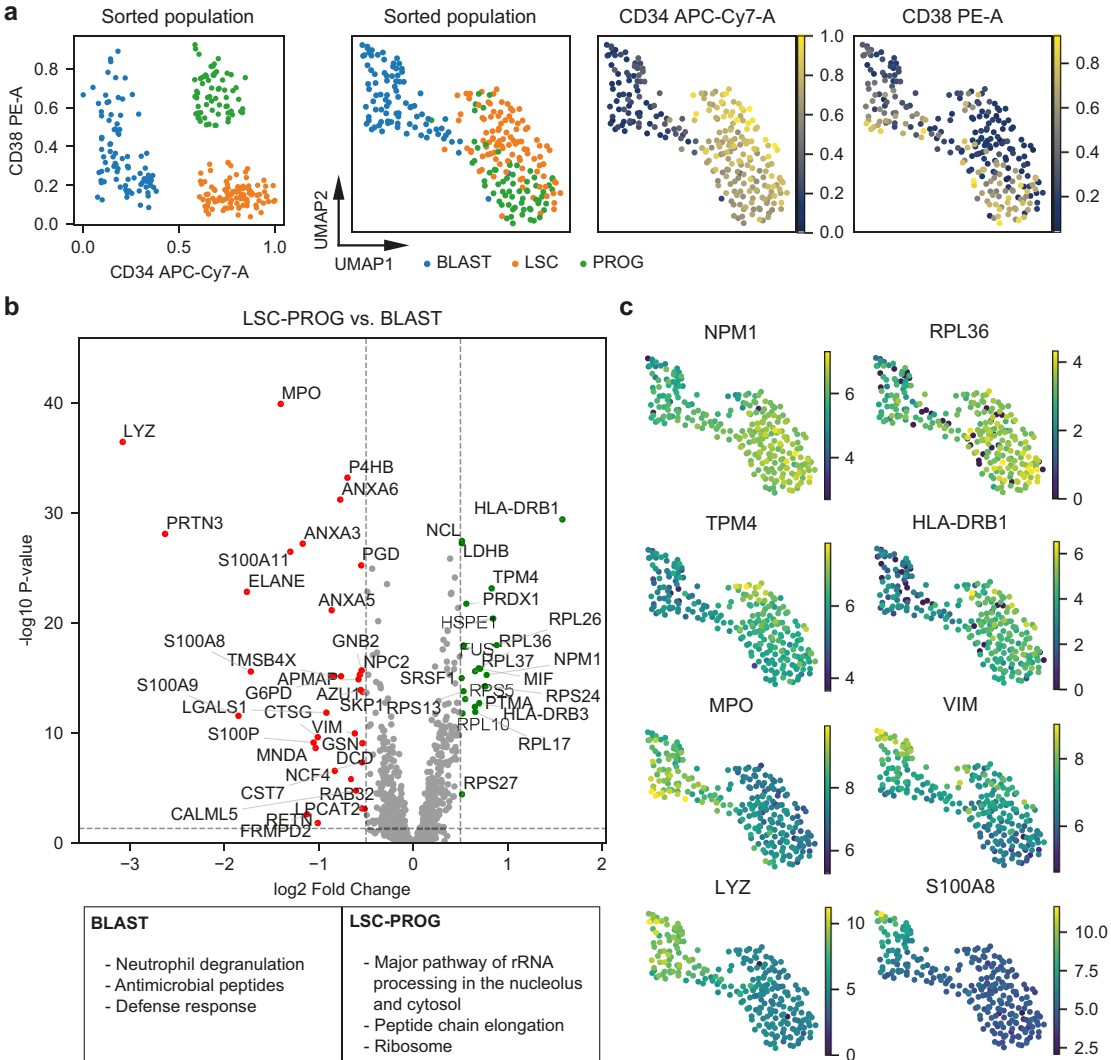

**Fig. 4 Extracting biological information from scMS data. a** FACS and scMS data from the 'high' dataset (255 cells). Left scatter plot shows the FACS derived expression of CD34 and CD38 of each cell, colored by differentiation stage annotation (BLAST = blast, LSC = leukemia stem cells, PROG = progenitors). Plots to the right show the UMAP embedding of cells using scMS data (255 cells, 1134 proteins), overlaid with differentiation stage annotation and FACS derived expression of CD34 and CD38. **b** Volcano plot of differential protein expression between cells labeled as LSC & progenitor and blast. Dashed horizontal line marks the significance threshold of 0.05 and dashed vertical lines mark the effect size threshold of an absolute log2 fold change of 0.5. Dots represent identified proteins, with up-regulated proteins marked as green and down-regulated proteins marked as red. The box at the bottom shows significantly enriched gene terms from the differentially expressed proteins in blasts and LSC & progenitor. **c** UMAP embedding of cells using scMS data, overlaid with scMS-derived protein expression of selected proteins. Source data are provided as a Source Data file.

In total, we processed eight bulk sorted single-cell plates (Supplementary Table 1), three with 1:1:1 booster and five with bulk booster, resulting in a dataset of 2723 proteins across 2025 cells with an average of 987 proteins detected per cell (Supplementary Table 2, Supplementary Fig. 7a). SCeptre successfully removed all empty wells from the analysis, indicating the validity of the filtering based on summed intensity (Supplementary Fig. 7b). Furthermore, we found that our normalization procedure successfully integrated all plates without the use of a reference channel, irrespectively of the different booster types used for these plates (1:1:1 & bulk) (Supplementary Fig. 7c). To annotate the cells with their respective differentiation stages, we gated the index FACS data in FlowJo and decided to further subdivide the blasts into CD38 + and CD38− populations (Supplementary Fig. 7d). As we are now sampling cells from the entirety of the OCI-AML8227 model, the main fraction of cells were blasts (1467 CD38−, 467 CD38 +) while also sampling 76 LSCs and 15 progenitors. As we would expect the dataset to

contain cells with intermediate differentiation stages, we used a diffusion map[51] to embed the protein expression data in 2-d space. The diffusion map revealed an ordering of cells in striking accordance with the FACS data with LSCs at the apex, giving rise to progenitors and blasts (Fig. 5a, Supplementary Fig. 8a). Surprisingly, we found both CD38 + and CD38− blasts in proximity to the LSCs and a slight separation of these two types of blasts along the trajectory, which seemed to terminate in CD38− blasts. This raised the possibility of the existence of two parallel differentiation paths towards terminal CD38− blasts rather than only the classical sequential 'LSC_progenitor_blast' path. This was further exemplified when we computed the diffusion pseudotime[52] from the protein data for each cell and observed overlapping cells in the CD38− compartment (LSC & CD38− blast) as well the CD38 + compartment (progenitors & CD38 + blast) (Fig. 5b). To further investigate what drives the observed cell ordering, we plotted protein expression across cells ordered by pseudotime (Fig. 5c).

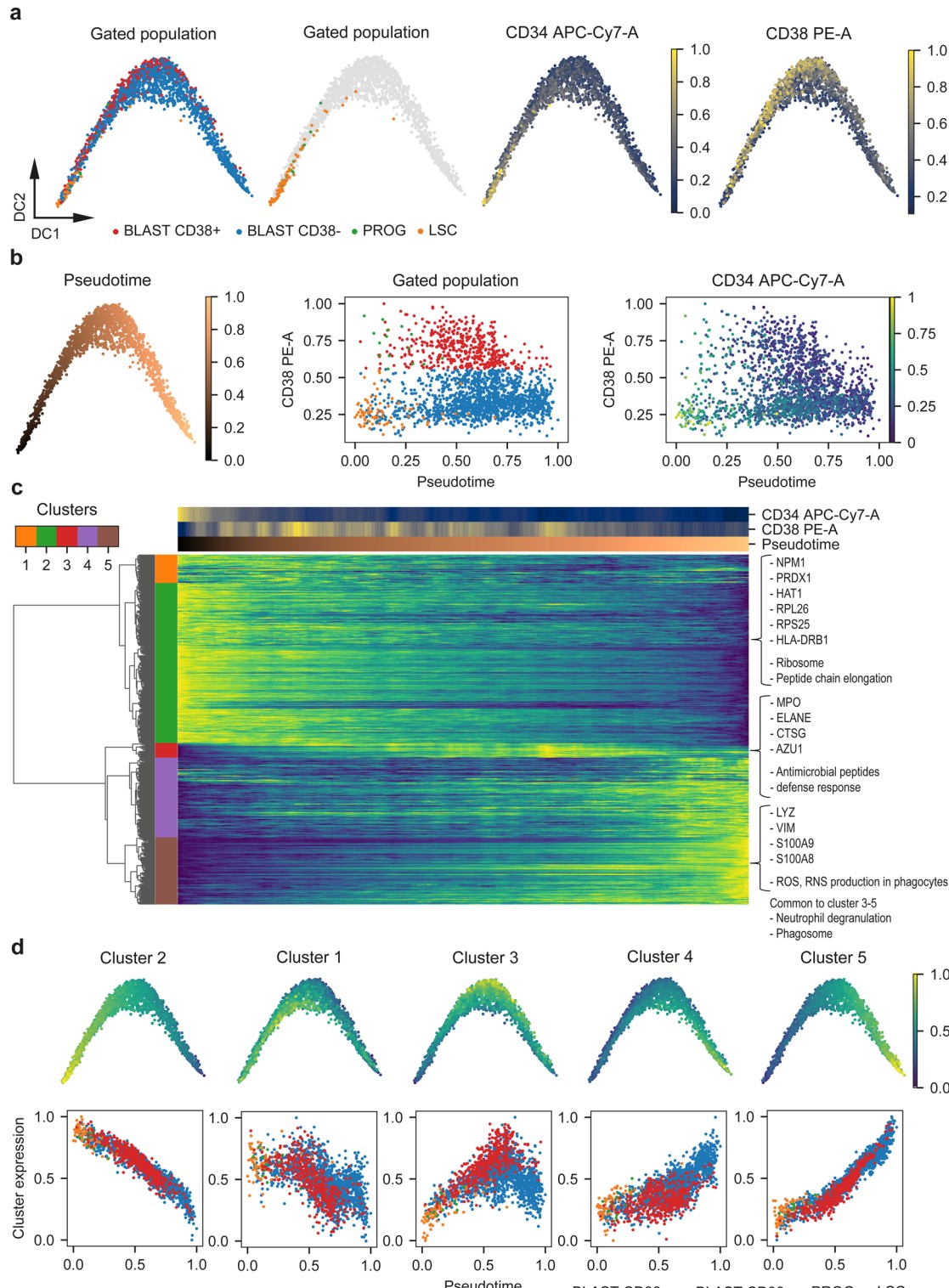

**Fig. 5 scMS recapitulates differentiation trajectories. a** Diffusion map based on imputed scMS data (2,025 cells, 2,723 proteins) overlaid with FACS derived cell gating and CD34 and CD38 expression. BLAST = blasts, PROG = progenitors, LSC = leukemia stem cells. **b** Left: Diffusion map overlaid with pseudotime, calculated using the scMS data. Middle: Scatterplot of cells with their calculated pseudotime and FACS derived CD38 expression, annotated with their gating (middle) or CD34 expression (left). **c** Heatmap of cells in the columns ordered in pseudotime and 479 selected proteins (Methods) in the rows. Proteins were clustered hierarchically into five clusters. Imputed protein expression values, CD34, CD38 and pseudotime for the ordered cells were smoothed by applying a moving average across 50 cells. Protein expression is normalized between 0 and 1. **d** Expression values of all proteins in each cluster were aggregated to a signature by taking the mean and normalizing between 0 and 1. Top: Signatures are plotted on top of the diffusion map. Bottom: Scatterplot of cells with their pseudotime and the signature of each cluster, annotated with their gating. Source data are provided as a Source Data file.

Hierarchical clustering of proteins revealed coordinated expression changes along the trajectory. Subsequently, we aggregated the proteins within each cluster into a mean expression value, and superimposed these signatures onto the diffusion map (Fig. 5d, Supplementary Fig. 8b). This revealed not only specific protein expression patterns in both LSC and terminal blast compartments, but also suggested that high expression of cluster 3 proteins seems to correspond to CD38 positivity, suggesting that this cluster is more strongly associated with the CD34−CD38 + blast compartment. In line with our earlier findings (Fig. 4), proteins of the early cluster 2 were enriched in gene terms associated with translation (Ribosome, Peptide chain elongation, Major pathway of rRNA processing in the nucleolus and cytosol) (Supplementary Data 2) suggesting that LSCs, like hematopoietic stem cells (HSCs)[53], are endowed with high protein synthesis activity. Moreover, we also noted that factors such as PRDX1 and HAT1, which are strongly associated with poor outcome in AML[54], were also found in cluster 2, in line with their enrichment in the LSC compartment (Supplementary Fig. 9a). Clusters 3-5 represent more mature differentiation stages as supported by their enrichment of myeloid-associated gene terms (Neutrophil degranulation, Phagosome). Consistent with the pseudotime analysis, the small cluster 3 is highly enriched in proteins of primary granula (including

ELANE, CTSG, MPO, AZU1), which are deposited early during granulocytic differentiation[55], and cluster 3 is consequently enriched for gene terms associated with this early granula (Antimicrobial peptides, Defense response) (Supplementary Fig. 9b). In contrast, proteins associated with clusters 4–5 display enrichment of terms representing functional neutrophils (such as ROS, RNS production in phagocytes) supporting their placement later in the differentiation hierarchy. Collectively, this analysis demonstrates the power of scMS in arranging cells along a differentiation trajectory and suggests the possibility of the existence of two parallel differentiation paths. Specifically, our data is compatible with both the conventional 'LSC_Progenitor_ CD38 + blasts_CD38− blasts' path and an unconventional 'LSC_CD38−' path that appears to bypass both the progenitor and CD34−CD38 + blast stages. To directly test the possibility of the existence of two distinct differentiation pathways for LSCs towards mature CD34-CD38− blasts, we sorted LSCs and progenitors and assessed their differentiation in culture over a 10-day period (Fig. 6). Consistent with the conventional path, progenitors initially generated CD34-CD38 + blasts, which further differentiated to CD34-CD38− blasts. Conversely, early differentiation (d2) of LSCs were clearly towards CD34-CD38− blasts before the later formation of progenitors (d4) and ultimate regeneration of

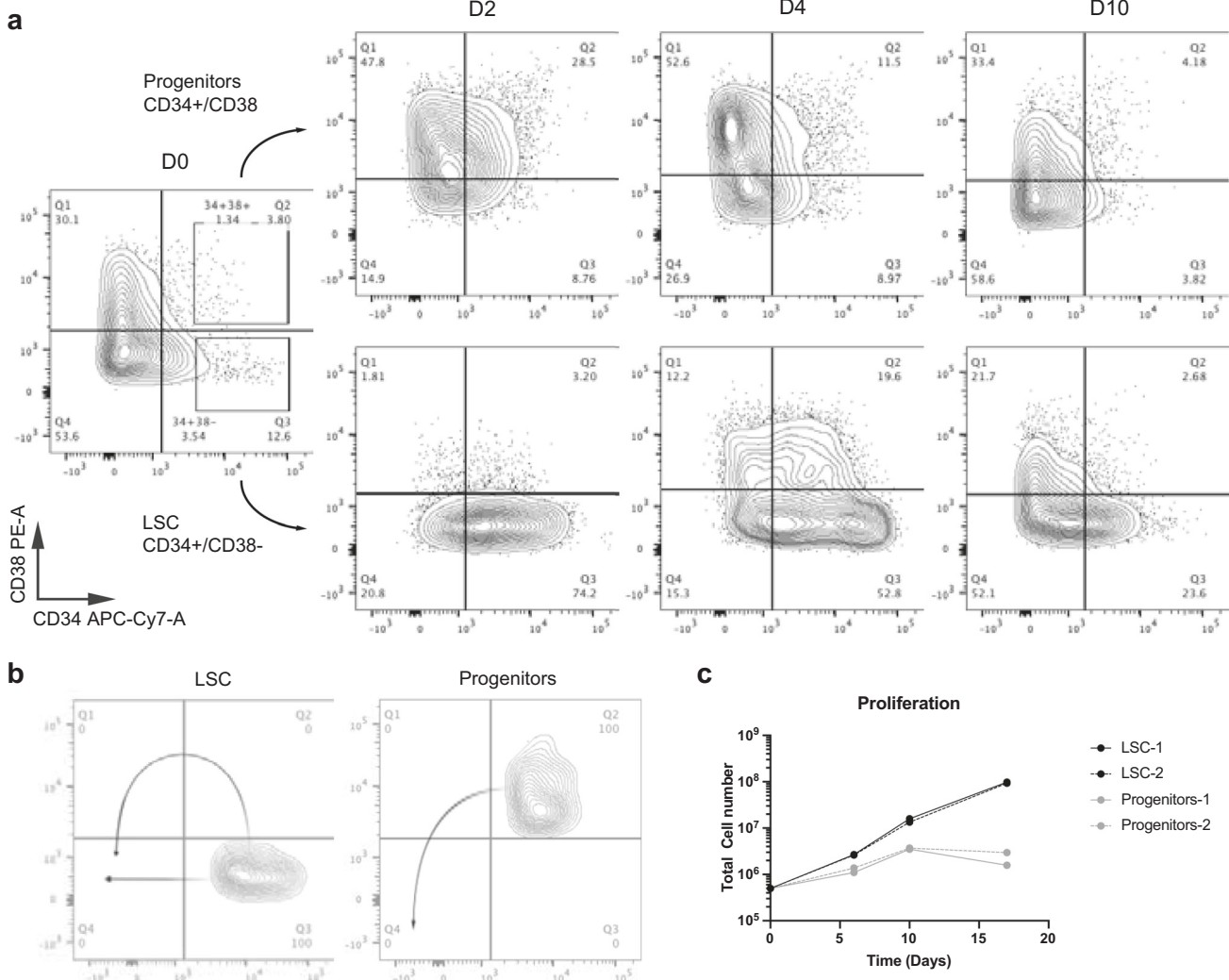

**Fig. 6 FACS differentiation assay of OCI-AML8227 culture system. a** Leukemia stem cells (LSCs) and progenitors were sorted and their differentiation was assessed over a 10-day period using FACS analysis. **b** Proposed differentiation of LSCs and progenitors. **c** Cell proliferation of cultures initiated by LSCs or progenitors as biological duplicates. Source data are provided as a Source Data file.

the entire culture (d10). Hence, these findings support the possible existence of two distinct differentiation pathways for LSCs and demonstrate the power of scMS analysis to uncover new biological mechanisms.

**Integration of unbalanced scMS datasets**. At this point, we have demonstrated our ability to detect heterogeneity in complex cell systems and that our normalization method effectively integrates samples that contain comparable cell numbers of each population. However, the latter would not be the case for the integration of the pre-enriched 'high' samples with the bulk samples, because the assumption of equal medians for each protein across samples would not be met. Nevertheless, we would ideally be able to integrate these two types of datasets, as the bulk approach is limited when sampling rare populations like e.g. the progenitors. Integration of heterogeneous datasets is a well-studied problem in scRNA-seq and we identified Scanorama[56] as a promising tool for unbiased i.e. unsupervised batch correction of unbalanced multiplexed scMS datasets. Scanorama detects mutual cell types across dataset and uses them to align the datasets to each other. To further increase the number of LSCs and progenitors, we measured an additional 384-well plate with pre-enriched populations to supplement the 'high' dataset (Supplementary Table 1). Subsequently, we used SCeptre to integrate the two 'high', pre-enriched plates into one dataset, and Scanorama to integrate this with the 'bulk' non-enriched dataset. This resulted in a dataset of 917 proteins across 2514 cells, consisting of 1586 CD38− blasts, 488 CD38 + blasts, 259 LSCs, and 181 progenitors (Supplementary Fig. 10a, b). Integration of additional proteins through more extensive imputation resulted in batch effects, indicating that the application of this tool to multiplexed scMS data should be evaluated on a case-by-case basis. The diffusion map shows similar results as in the bulk dataset, with LSCs giving rise to progenitors, followed by a mix of CD38 + and CD38− blasts (Fig. 7). Nonetheless, the increased number of LSCs and progenitors strengthened the evidence for the differentiation of LSCs into progenitors. Furthermore, similar protein expression signatures were identified, thereby supporting the biological relevance of these integrated analyses (Supplementary Fig. 10c, d, Supplementary Data 2).

## Discussion

This work represents a proof-of-concept study, investigating to what extent current technology and subsequent computational workflows are able to conduct semi high-throughput single-cell proteomics analysis on a relevant AML model system. By spending considerable effort not only on the sample preparation and data generation methods, but also on the subsequent data analysis, we managed to establish a scMS workflow which is able to (1) quantify approximately 1000 proteins per cell, (2) analyze more than a hundred cells per day of instrument time, (3) normalize, filter, integrate and visualize the data using state-of-the-art single-cell computational algorithms, and (4) detect heterogeneity and cell-specific proteins which may serve as a starting point for further investigation of yet undiscovered cell states and potential therapeutic targets or other functionally relevant candidates.

Single-cell approaches put significant strain on throughput requirements; usually a high number of cells is required to gain confidence in biological observations or to identify rare sub-populations. Therefore, we focused especially on having an experimental workflow that would allow easy preparation of large cell numbers and demonstrate the analysis of nearly 3000 single cells in a limited timeframe. By using standard FACS sorting methodology, we were able to sort several 384-well plates per hour and allow for including FACS-based index information in the data analysis. This allows links to be drawn between fluorescent surface markers and expression levels of detected proteins within the cells. This feature is especially relevant for complex in vivo systems such as primary bone marrow, where a multitude of relevant surface markers, to which significant biological properties have already been assigned, can be integrated.

In line with our overall goal of using standardized lab consumables, we also used conventional chromatography setups to ensure the adaptation of the workflow in labs across the world. By including a trap column, we added robustness, which means that we in general are able to run more than two weeks' worth of samples on the same analytical column (>1,500 single cells). In terms of reagents and consumable costs, excluding the MS instrument acquisition, our workflow expenses are estimated to ~1USD per cell and thus provide a realistic means of conducting large-scale scMS studies.

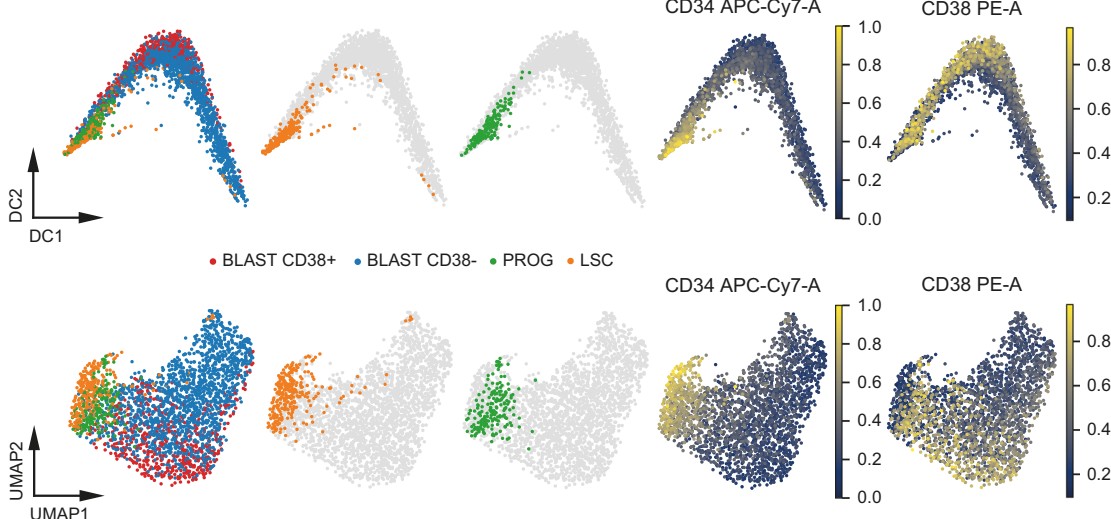

**Fig. 7 Integration of unbalanced scMS datasets.** Diffusion map and UMAP embedding of integrated dataset (2514 cells, 917 proteins), overlaid with the FACS derived gated populations, CD34 and CD38 expression. BLAST = blasts, PROG = progenitors, LSC = leukemia stem cells. Source data are provided as a Source Data file.

Compared to the sole other publicly available multiplexed scMS experimental pipeline using common laboratory consumables, SCoPE2, our workflow distinguishes itself on a few key points. Besides using TFE as lysis buffer instead of water, we also opted not to dedicate two TMT channels to a reference sample and an empty well, as the normalization procedure implemented in SCeptre was sufficient to correct for any observable batch effect, and the QC part of the workflow ensured that only data originating from single-cell wells was used. This results in a slightly increased multiplexing capacity of 14 single cells per run. However, the previously observed decrease of quantification performance in the channels adjacent to the booster channel[35] should be considered in future experimental designs, especially when boosting is increased beyond 200x. This could be the case for e.g. smaller cells, where more peptide signal is required to identify a sufficient number of proteins. In general, the amount of boosting should be carefully evaluated in each experimental setting in terms of its impact on the single-cell quantitative accuracy[35,36]. Finally, signal from empty wells should be specifically determined for each experiment.

We demonstrated that with a randomized TMT labeling layout, the simple normalization strategy included in SCeptre successfully integrates balanced experimental designs. Further, we identified Scanorama as a potential integration tool for unbalanced designs. As shown here, this opens up the possibility to enrich for rare populations in addition to the unbiased bulk-fashion, and subsequently analyze all cells in unison. However, we note that Scanorama only succeeded in integrating a subset of high-coverage proteins, thus highlighting an interesting computational challenge. Furthermore, Scanorama requires overlapping cell types across datasets, which should be considered in the experimental design. Hence, alternative normalization strategies for multiplexed single-cell proteomics should be readily adapted by the scMS field.

We showed that the frequently used k-nearest-neighbors imputation can be used on our scMS data and that it can improve separation of the differentiation stages. However, we note that imputation should be carefully evaluated in terms of batch effects and the effect on cell separation, and that this evaluation might be complicated by a lack of accurate cell labels. Nonetheless, as seen in the scRNAseq field, scMS should benefit from imputation strategies that consider the specific nature of missing values and noise in isobaric quantification.

In terms of extracting biological meaning from scMS data, it has to be underlined that this work, for the first time demonstrates the ability to separate cell differentiation stages from the same exact starting pool of cells, rather than using separate pools as starting material for the FACS sort[25,28]. Moreover, the similarity of AML cells within the hierarchy, and especially between the LSC/Progenitor populations, requires accurate and sensitive protein measurements and sufficient proteome coverage in order to be able to separate them in UMAP space. This is also exemplified by the fact that the 'high' dataset was slightly better able to resolve these two populations compared to the 'medium' data, again underlining the importance of high ion count measurements for accuracy. However, balancing proteome depth and quantification accuracy will remain challenging even when sensitivity of scMS increases and should be evaluated in every specific experimental setting. Especially in the context of identifying cell stage-specific proteins, it has to be evaluated how many cells will need to be sampled in order to reach a meaningful number of cells for specific biological questions. Nevertheless, by deploying our workflow on >2,000 single cells sorted as bulk (i.e. non-enriched), we are able to separate the cell differentiation stages, interrogate their differentiation trajectories and discover cell-specific functional differences as highlighted by GO-term enrichment and diffusion map trajectories.

In conclusion, this work presents the first single-cell analysis of a leukemia hierarchy using an LC–MS-based proteomics method. We demonstrate that within our model AML system, we are able to recapitulate the FACS data using our proteomics data. While we focus on a single patient leukemia, it should nevertheless be a good resource for follow-up studies, and paving the way for studying primary leukemia and other blood-related disorders using scMS. We build on the original excitement inspired by ScoPE-MS, and hope this method continues to open up a plethora of research avenues, spanning across many biological fields. Proteome coverage will only improve as instrument sensitivity and experimental workflows develop further, closing the current coverage gap between RNA-based and protein-based approaches that still exists at single-cell level, with great strides already having been made in the first few years of a field still in its infancy.

## Methods

**Cell culture and FACS sorting.** OCI-AML8227 cells were grown in StemSpan SFEM II media, supplemented with growth factors (Miltenyi Biotec, IL-3, IL-6 and G-CSF (10 ng/mL), h-SCF and FLt3-L (50 ng/mL), and TPO (25 ng/mL) to support the hierarchical nature of the leukemia hierarchy captured within the cell culture system. On day 6, cells were harvested (8e6 cells total), washed, counted, and resuspended in fresh StemSpan SFEM II media on ice at a cell density of 5e6 cells/ml. Staining was done for 30 mins on ice, using a CD34 antibody (CD34-APC-Cy7, Biolegend, clone 581) at 1:100 (vol/vol) and CD38 antibody (CD38-PE, BD, clone HB7) at 1:50 (vol/vol). Cells were washed with extra StemSpan SFEM II media, and subsequently underwent three washes with ice cold PBS to remove any remaining growth factors or other contaminants from the growth media. Cells were resuspended for FACS sorting in fresh, ice cold PBS at 2e6 cells/ml. Cell sorting was done on a FACSAria I or III instrument, controlled by the DIVA software package (v.8.0.2) and operating with a 100 µm nozzle (Supplementary Fig. 11). Cells were sorted at single-cell resolution, into a 384-well Eppendorf LoBind PCR plate (Eppendorf AG) containing 1 µl of our custom lysis buffer (80 mM Triethyl-lammonium bicarbonate (TEAB) pH 8.5, 20% 2,2,2-Trifluoroethanol (TFE), 10 mM tris(2-carboxyethyl) phosphine (TCEP) and 40 mM Chloroacetamide (CAA)). Directly after sorting, plates were briefly spun, snap-frozen on dry ice, and then boiled at 95 °C in a PCR machine (Applied Biosystems Veriti 384-well) for 5 mins. Plates were again snap-frozen on dry ice and stored at -80 °C until further sample preparation. The same procedure was followed for the booster plates and the plate for the MS3 bulk data, but instead of sorting single cells, 500 cells were sorted in three-way purity into each well, containing 1 µl of lysis buffer. For the assessment of LSC and progenitor differentiation trajectories, 50,000 cells (FACS sorted as above) were used to initiate replicate cultures in 100 µL of media supplemented with cytokines. Cells were harvested after 2, 4, 6 days, stained with antibodies and analyzed by flow cytometry as above (BD Aria III). At Day 6, cells were also counted and re-seeded at 50,000 cells/100 µL. Profiles and proliferation were reassessed again at Day 10 and Day 17.

**Mass spectrometry sample preparation of single-cell samples.** After thawing, protein lysates from the single cells were digested with 2 ng of Trypsin (Sigma cat. nr. T6567), dissolved in 1 µl of 100 mM TEAB pH 8.5 containing Benzonase (Sigma cat. nr. E1014) diluted 1:500 (vol/vol) to digest any DNA that would interfere with downstream processing. For the booster plates, the amount of trypsin was increased to 10 ng in order to digest the protein content of each well containing 500 cells. Plates were vortexed and kept at 37 °C overnight to complete the protein digestion. All dispensing steps in this protocol were done using the Dispendix I-DOT One instrument. After digestion, peptides were labeled with TMTPro reagents. Some example labeling layouts can be found in Supplementary Data 1. 6 mM of each label was added to the single-cell wells, while the 500-cell booster plate wells were labeled with 13 mM of TMTPro-126 reagent (in 1 µl volume) in each well or in case of the reference channel with TMTPro-127N. Subsequently, plates were kept at RT for 1 h. The labeling reaction was quenched with 1 µl of 2.5% Hydroxylamine for 15 min, after which peptides were acidified with 1 µl of 2% TFA. For the empty wells, all steps were the same except no cells were sorted into the wells. Subsequently, the booster plates were pooled according to cell differentiation stage (blasts, progenitors and LSC), and desalted as five (including two reference channels: 1:1:1 & bulk) individual pools using a SOLAµ C18 96-well plate. Eluted and desalted peptides were concentrated to dryness in an Eppendorf Speedvac, after which they were resuspended in 1% TFA, mixed 1:1:1 in case of the 1:1:1 booster and diluted to a final concentration of the peptide equivalent of 35 cells/µl. Final TMT samples were then mixed from 14 or 13 single cells plus the equivalent of 200 booster channel cells and in case of the reference channel plus the equivalent of 10 cells. This pooling was performed using the Opentrons OT-2 liquid handler, except for the 'medium' and 'high' plates, which were pooled with a handheld multichannel pipette. The resulting peptide mix was concentrated once more in an Eppendorf Speedvac, and re-constituted in 1% TFA, 2% Acetonitrile,

containing iRT peptides (Biognosys AG, Switzerland) for individual Mass Spectrometry (MS) analysis.

**Mass spectrometry sample preparation of MS3 bulk data.** The plate for the MS3 bulk data was treated the same as a booster plate, with the exception that it was labeled so that an equal number of wells of each differentiation stage were labeled with the following channels: blast—127 N, 131 N, 132 C; progenitor—129 N, 131 C, 133 N; LSC—128 N, 130 C, 133 C. This resulted in 60,000 cells being labeled for each differentiation stage (20,000 cells per channel in triplicate). All wells of the plate were pooled together into one aliquot, and desalted using a SOLAμ C18 96-well plate. Eluted and desalted peptides were concentrated to dryness in an Eppendorf Speedvac, after which they were resuspended in high-pH buffer (5 mM ammonium bicarbonate, pH 10). 20 μg of peptides were fractionated using an offline Thermo Fisher Ultimate3000 liquid chromatography system at a flowrate of 5 μl/min over a 60 min gradient (from 5% to 35% acetonitrile), while collecting fractions every 2 min. The resulting 20 fractions were pooled into 10 final fractions (fraction 1 + 11, 2 + 12, etc.), concentrated to dryness, and re-constituted in 1% TFA, 2% Acetonitrile for individual Mass Spectrometry (MS) analysis.

**Mass spectrometry data collection of single-cell samples.** Peptides were loaded onto a 2 cm C18 trap column (ThermoFisher 164705), connected in-line to a 15 cm C18 reverse-phase analytical column (ThermoFisher EasySpray ES804A), with 100% Buffer A (0.1% Formic acid in water) at 750 bar using the ThermoFisher EasyLC 1200, and the column oven operating at 30 °C. Peptides were eluted over a 160-min gradient at a flowrate of 100 nl/min, using 80% Acetonitrile, 0.1% Formic acid (Buffer B) going from 8% to 23% over 88 min, to 38% over 42 min, then to 60% over 10 min and to 95% over 5 min, and holding it at 95% for 15 min. Spectra were acquired with an Orbitrap Exploris™ 480 Mass Spectrometer (ThermoFisher Scientific) running Tune 1.1 and 2.0SP1 with Xcalibur 4.3, operating in DD-MS2 mode, with FAIMS Pro™ Interface (ThermoFisher Scientific) cycling between CVs of -50 V and -70 V every 1.5 s. MS1 spectra were acquired at 60,000 resolution with a scan range from 375 to 1500 m/z, normalized AGC target of 300% and maximum injection time of 50 ms. MS1 precursors with an intensity > 1.0e4, charge state of 2–6, and that matched a precursor envelope fit threshold of 70% at 0.7 m/z fit window were selected for MS2 analysis. Here, ions were isolated in the quadrupole with a 0.7 m/z window, collected to a normalized AGC target of either 150%, 300%, or 500% or maximum injection time (IT) of 150 ms, 300 ms, 500 ms, or 1000 ms, fragmented with 32 normalized HCD collision energy and resulting spectra acquired at 45,000 resolution with a first mass of 110 m/z to ensure appropriate coverage of the TMTPro reporter ions. Precursors that were sequenced once were put on an exclusion list for 120 s, exclusion lists were shared between CV values and Advanced Peak Determination was set to 'off'. The scMS samples for the comparison of 20% TFE versus LC–MS grade water as lysis buffer were analyzed on a Q-Exactive HF-X, operating at 60,000 resolution at MS1, with an AGC target of 3e6 and max IT of 50 ms. Precursors with an intensity greater than 2e4 were selected for MS2 analysis as a 'top13' method, isolated at 1.4 m/z isolation width, collected at a maximum IT of 150 ms or AGC target of 1e5, then fragmented with a normalized collision energy of 30 and resulting spectra analyzed in the Orbitrap at a resolution of 45,000 and first mass set to 110 m/z. Here, peptides were eluted over a 120-min gradient at 100 nl/min, going from 8 to 23% over 60 min, to 38% over 30 min and to 60% over 10 min, after which 95% buffer B was reached in 5 min and held for 15 min.

**Mass spectrometry data collection of MS3 bulk data.** Peptides were loaded onto a μPAC™ trapping column (PharmaFluidics), connected in-line to a 50 cm μPAC$^{TM}$ analytical column (PharmaFluidics), with 100% Buffer A (0.1% Formic acid in water) using the UltiMate™ 3000 RSLCnano System (ThermoFisher), and the column oven operating at 45 °C. Peptides were eluted over a 140-min gradient at a flowrate of 250 nl/min, using 80% Acetonitrile, 0.1% Formic acid (Buffer B) going from 4% to 23% over 66 min, to 38% over 30 min, then to 60% over 10 min and to 95% over 4 min, and holding it at 95% for 9 min. Spectra were acquired with an Orbitrap Fusion™ Tribrid™ Mass Spectrometer (ThermoFisher Scientific) running Tune 3.4, operating in DD-SPS-MS3 mode, with FAIMS Pro™ Interface (ThermoFisher Scientific) cycling between CVs of -40 V, -60 V, and -80 CV every 2 s. MS1 spectra were acquired at 120,000 resolution with a scan range from 400 to 1600 m/z, normalized AGC target of 100% and maximum injection time on auto. To filter MS1 precursors, monoisotopic peak determination was set to peptide, intensity threshold to 5.0e3, charge state to 2–6 and dynamic exclusion to 60 s with the single charge state option activated. Precursor ions were isolated in the quadrupole with a 0.7 m/z window, collected to a normalized AGC target of 100% or maximum injection time (auto), and subsequently fragmented with 32 normalized CID collision energy. Spectra were acquired in the ion trap in turbo mode. For the MS3 spectrum, the 10 most intense fragment ions were selected from the MS2 spectrum after filtering out the precursor ion with 50 m/z below and 5 m/z above the peak and excluding isobaric tag losses of TMTpro. The precursor ion for the MS3 scan was isolated from the MS1 scan with a 0.7 m/z window collected to a normalized AGC target of 250% or maximum injection time (auto), and subsequently fragmented as defined previously. Previously selected fragments were isolated synchronously and fragmented again with 60 normalized HCD collision

energy. MS3 spectra were acquired in the Orbitrap with 50,000 resolution in a scan range of 100–500 m/z.

**Mass spectrometry raw data analysis.** Single-cell raw files were analyzed with Proteome Discoverer 2.4 (ThermoFisher Scientific) with the built-in TMTPro Reporter ion quantification workflows. Default settings were applied, with Trypsin as enzyme specificity. Spectra were matched against the 9606 *homo sapiens* database obtained from Uniprot (Swiss-Prot with isoforms, downloaded on 07/11/2020). Dynamic modifications were set as Oxidation (M), and Acetyl on protein N-termini. Cysteine carbamidomethyl was set as a static modification, together with the TMTPro tag on both peptide N-termini and K residues. Spectra were searched using the Sequest search engine and validated with Percolator. All results were filtered to a 1% FDR. For the reporter ion quantification, normalization mode and scaling mode were set to None and average reporter s/n threshold was set to 0. Isotopic error correction was applied. MS3 bulk data was analyzed using the standard settings of the Tribrid TMTpro SPS MS3 workflow using Sequest and Percolator and the Reporter Quantification Consensus workflow. Log2-fold-changes were calculated by Proteome Discoverer from the three technical replicates (three reporter channels) per differentiation stage.

**Data analysis for the comparison of different IT settings.** Technical triplicates of each of the four methods were first normalized by equalizing the median s/n of the pooled single-cell channels for each protein across replicates (one correction factor per protein). For each method, for each protein, 14 CV values were calculated from the 3 replicated measurements (i.e. 14 pooled single-cell channels). Only proteins with that were quantified in all 3 replicates were considered for CV calculation. CV was calculated by dividing the standard deviation of the normalized s/n of each protein in each single-cell channel across replicates by the mean of the raw s/n for each protein in each single-cell channel across replicates. To report a mean CV for each protein in each method, the mean of up to 14 CV values was calculated. To report a mean log2 s/n for each protein in each method, the mean of all s/n values of a protein in all 3 replicates was log2 transformed. To calculate the silhouette coefficient, the normalized data were further processed by a median shift across cells, log2 transformation and removal of either all proteins containing missing values across all methods, or across the individual method. Subsequently, protein data was transformed into the 20 first principal components (PCs). The silhouette coefficients of blast and LSCs were calculated per method using the 'silhouette_samples' function from Scikit-learn[57], providing the differentiation stage as labels and the first 20 PCs as matrix.

**Computational analysis of single-cell data.** FACS.fcs files were processed in FlowJo 10.7.1 with the IndexSort 2.7 plugin to apply bi-exponential transform and to gate the cells. FACS data and sort- and label layouts were used to create the metadata for each cell. Metadata and the Proteome Discoverer protein table were loaded into a Scanpy AnnData object. Analysis was performed with python 3.7.9 and Scanpy version 1.6.1.dev102 + g8d9eec4c. FACS data were normalized between zero and one. Potential contaminant proteins were removed. Failed raw files were removed and for the 'high' plate, rows I and J were removed. The SCeptre normalization method equalized the median protein expression values across files and channels in an iterative manner[58] until the highest change in the matrix compared to the previous matrix was below s/n of 1.1, which is slightly above the noise level. Missing values were ignored during the calculations. After normalization, values below 1.1 were set to missing. To remove outlier cells, upper and lower bounds for the log2 sum s/n were determined by MAD-based outlier detection and a minimum protein per cell cutoff was applied. Proteins quantified in less than 3 cells were removed. Subsequently, a median shift of total intensity across cells was applied to normalize for cell size and sampling depth, and finally, data were log2 transformed. Before imputation, the optimal threshold to remove low coverage proteins was determined by testing many thresholds with the extremes being (i) the removal of all proteins with any missing value, and (ii) keeping all proteins. The optimal threshold was selected based on the separation of differentiation stages measured as silhouette score (mean of all silhouette coefficients) in UMAP space (first 2 dimension). Data were imputed using the k-nearest neighbors method with 5 nearest neighbors. Subsequently, protein expression was scaled to unit variance and zero mean. To embed the protein data, principal component analysis (PCA) was performed, from which a cell-neighborhood graph was calculated. The neighborhood graph was embedded using UMAP[59], Force-directed graph drawing[60] or a diffusion map[51]. For the analysis of differential protein expression between two groups, the previously stored normalized log2 transformed protein expression matrix was used. For each protein, a two-sided Welch's t-test was performed, excluding missing values. P-values were corrected via the Benjamini–Hochberg procedure and a cutoff of 5% FDR was applied. All protein log2 fold-changes were calculated using the normalized raw data. For the analysis of enriched terms, the protein annotations provided by Proteome Discoverer were used, a hypergeometric test was performed, p-values were corrected via the Benjamini–Hochberg procedure and a cutoff of 5% FDR was applied. For the heatmap analysis in Fig. 5, a subset of 479 proteins was selected that showed changes across the trajectory by performing leiden clustering[61] on the cell-neighborhood graph and subsequently performing a differential expression test on

each cell cluster against all other cells, using the normalized but non-imputed raw data. Proteins were filtered to be detected in at least 200 cells and to have a log2 fold-change of at least 0.15 at a significance level below 0.05. Using all proteins for the analysis lead to similar results. For the integration of unbalanced datasets, the 'high' and 'enriched' datasets were processed together in SCeptre as described. Before imputation, only proteins were retained that had at least 40% valid values across cells. The same threshold was applied on the 'bulk' dataset. This resulted in 917 proteins overlapping between both datasets. Subsequently, Scanorama was applied with standard settings. All proteins were used for the embedding. For the heatmap, a subset of 481 proteins was selected that showed changes across the trajectory by performing leiden clustering on the cell-neighborhood graph and subsequently performing a differential expression test on each cell cluster against all other cells, using the imputed, normalized, and scaled data. Proteins were filtered to have a fold-change of at least 1.03 at a significance level below 0.05.

**Reporting summary**. Further information on research design is available in the Nature Research Reporting Summary linked to this article.

## Data availability
The mass spectrometry data have been deposited to the ProteomeXchange Consortium (http://proteomecentral.proteomexchange.org) via the PRIDE partner repository[62] with the dataset identifier PXD020586. All data is also available from the corresponding authors, and all relevant source data are provided with this paper.

## Code availability
The SCeptre package is available with https://doi.org/10.5281/zenodo.4631707[63] under the MIT license on Github (www.github.com/bfurtwa/SCeptre), containing the complete analysis performed in this study as multiple Jupyter notebooks.

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

## Acknowledgements

Erwin M. Schoof is a former Lundbeck Fellow (fellowship # 2017-389) and EMBO Fellow (ALTF 1595–2014) co-funded by the European Commission (LTFCOFUND2013, GA-2013-609409) and Marie Curie Actions. This work was supported by the Kirsten & Freddy Johansen Foundation, the Independent Research Fund Denmark, the Svend Andersen Foundation, the Candys Foundation, and through a center grant from the Novo Nordisk Foundation (Novo Nordisk Foundation Centre for Stem Cell Biology, DanStem; Grant Number NNF17CC0027852). B. Furtwängler is the recipient of a fellowship from the Novo Nordisk Foundation as part of the Copenhagen Bioscience Ph.D. Programme, supported through grant NNF19SA0035442. U. auf dem Keller acknowledges support by a Novo Nordisk Foundation Young Investigator Award (Grant Number NNF16OC0020670). Work in the Dick lab was supported by funds from the: Kirsten & Freddy Johansen International Prize, Princess Margaret Cancer Centre Foundation, Ontario Institute for Cancer Research with funding from the Province of Ontario, Canadian Institutes for Health Research, Canadian Cancer Society Research Institute, Terry Fox Foundation, Genome Canada through the Ontario Genomics Institute, and a Canada Research Chair. We also acknowledge funding support from PRO-MS: Danish National Mass Spectrometry Platform for Functional Proteomics (grant no. 5072-00007B). Some figures were created with the help of BioRender.com. We thank the Vertical Marketing Team of ThermoFisher Scientific, San Jose, for input on instrument methods, and the BRIC FACS facility for support with the flow cytometry work.

## Author contributions

E.M.S., B.F., and N.Ü. designed and carried out experiments, wrote the manuscript, and conducted data analysis. B.F. and N.R. designed and implemented the computational workflow. E.L. and C.G. contributed with cellular assays. S.S. and U.a.d.K. assisted with the MS analysis, helped develop the instrument methods, and contributed to the overall functioning of the method. All authors proofread and contributed to the manuscript. U.ad.K., J.E.D., and B.T.P. oversaw the study, wrote the manuscript, and helped design the experiments.

## Competing interests

The authors declare no competing interests.
