## [Peer Review File · Nature Communications]

REVIEWER COMMENTS

Reviewer #1 (Remarks to the Author):

I will here address the point-by-point discussion provided by the authors.

1. Thank you for the clarification.
2. Thank you for taking the time to test Fido and Dart-ID. It is interesting, and possibly not surprising, to see that proteins with only unique peptide hits are rescued.
3. The authors have tested the effect of imputing some values, which is useful given the large number of missing values in single cell MS-based experiments. They assess the impact of imputation on these data for different filtering criteria (Fig 5b), using silhouette scores, which I find a very useful approach. I was however initially surprised not to see any silhouette plots in the data after the repeated mention of silhouettes, and it took me some time to understand that they used the mean silhouette score. Typically, scores are computed for each data point in the set to be clustered, and all these are then visualised on a plot. Scores can also be

averaged for each cluster to provide cluster-level validation metrics. These metrics and visualisations could certainly be made use of. The authors should at least clarify in the main text that they use the mean value (it does appear in the material and methods later), and ideally acknowledge that they then miss the variation within that score. See also below.

4. It is unclear how 5 relates to the additional GO/pathway enrichment analyses. In Fig 4, I see that the box under the volcano plot 2b is indeed relevant.

5 and 6. Thank you for making a Python package available. It would still benefit from some user-level documentation. Scanpy and scanorama, which the authors refer to, are good examples thereof.

7. I have looked at the `create_meta_data` function. It looks like the function, and with it SCEPTRE, are focused on Proteome Discoverer output files (see also line 201). If so, please mention this. Otherwise, please document how `create_meta_data` (or `load_dataset`) can be used more generally.

8. Overlay and annotate are spot on.

9. As discussed above, while using silhouette scores and plots is an useful approach, the implementation is a bit inelegant. I would suggest, at least, to visualise the distributions of the individual scores in Figure 3b, rather than the inadequate bar plots. The

comparison of silhouette score distributions also applies to supplementary figure 5b.

10. Thank you for the clarification.

11. Thank you for the update. I also appreciate the usage of scanorama. I am not sure that the claims of "unbiased removal of batch effect" are totally appropriate (lines 97 and 224). You have demonstrated batch removal, but doing so without (some) bias is unlikely and more difficult to demonstrate.

12. Thank you for the clarification and the rephrasing.

13. Thank you for your clarifications. I understand that your iterative procedure does remove unwanted technical/batch variances while keeping biological variance, but I have a hard time appreciating that it is a very good one. It would be interesting to compare it with other approaches, albeit this is probably out of the scope of this manuscript.

14. It is reassuring that both approaches return comparable results.

15. Thank you for the clarification.

16. I am not sure whether 5 nearest neighbours is any better than 2 for imputation. Ideally, the authors would apply a procedure as shown on their supplementary figure 5b with different values of k ,

thus formalising and validating their choice.

17 and 18. Thank you for the clarification. See my comments on silhouette scores/plots above.

20. As mentioned above, the README file should be completed with documentation of the package or at least a description/enumeration of its the functions. It could also mention the MIT license.

I got access to SCeptre through PRIDE; this is not an adequate way to distribute code. I assume this is a temporary means until the Github repository becomes available. Please remove the SCeptre.zip file before publication of the data and Github repository to avoid confusion; users are likely to use that static version and miss new features and bug fixes.

Other comments:

- Please update reference 25. SCoPE2 has now been published (<https://doi.org/10.1186/s13059-021-02267-5>).

- This is probably my limited expertise in biochemistry speaking for itself here, but it would be useful to define TFE which. Unless I missed it, it isn't defined.

- Supplementary tables 1 and 3 should be proper tables, not as pdf files.

Laurent Gatto

Reviewer #2 (Remarks to the Author):

The authors have produced a much improved manuscript – although I note it has now switched journals.

I am pleased that the authors have answered many of my concerns in their revised manuscript. One set of revisions I feel are still very misleading and this surrounds quantitative accuracy.

To address the issue of accuracy I think the authors carried out an experiment where they compared blasts versus LSC cells at the single cell level and also blasts versus LSCs using bulk cells in a typical TMT based proteomics workflow. Both experiments gave fold changes in abundance between the two sets of cells and the authors compared whether fold changes acquired in the single cell experiment with what was achieved in bulk cell experiment using Pearson correlation. The experimental design is poorly described in the text and I have no idea how the data presented in the figures listed below are related to one another. The explanation given on lines 181-183 was not clear to me.

The results of the comparisons, as far as I understand are presented as the following:

Figure 3 - Pearson correlation of the protein fold changes but only for the top 400 high-coverage proteins

Figure S3C – Pearson Correlation between fold changes for all injection times, but only proteins that have no missing values (top) or that are present in all experiment (bottom)

Figure S5C – Pearson correlation now just for 300ms and 500ms where non-imputed values were used and only proteins with $n \geq 3$ in blasts and LSCs respectively were considered

These data were also presented in figure 2d in a different form.

In the rebuttal letter the authors also note that there is a lower dynamic range of fold changes observed in single cell data than the bulk data and note improved accuracy – as judged by Pearson correlation scores – when longer inject times are used.

I think the data as presented are very misleading. Pearson correlation scores are not a substitute for accuracy. In simple terms, if a fold change of 2 is measured for a pair of proteins in the bulk data, is the fold change of the same pair of proteins 2 measured in the single cell data comparable? Showing

a linear correlation does not show that the same fold change values are achievable by both the bulk and single cell workflows. The Pearson correlation figures show a lot of scatter of data off the diagonal and no measure of the variability of the measurements across the intensity range of the measurements. I think the technical variability of the system is not fully explored here. Moreover, figure 3C in the main text portrays basically cherry picked data – i.e. the top 400 most intense measurements. Figure S5C shows a very different picture but is hidden in the supplementary data. I suggest that FigS5C should be promoted into the main body of the manuscript along with a fuller explanation of accuracy issues in the text.

The authors need to do a much better job depicting the accuracy of the single cell data. Additionally, in the main text they need to note the differences in dynamic range of fold change measurements and look for systematic bias in fold change measurements across the intensity range.

Thorough analysis of

A) Number of cells in the booster channel

B) FAIMS settings

Have been discounted as outside the scope of the manuscript and the authors seem happy to go with values suggested by other labs. In the case of the latter I agree with their sentiments. For the former however, every cell line may be different in terms of the amounts of protein recovered in the workflow, and 200 cells as a booster channel may be OK for one cell types but not for the next. This needs further exploration in the text.

Reviewer #3 (Remarks to the Author):

In their revised manuscript the authors substantially improved the study, and I now find it suitable for publication in Nature Communications.

In my view the addition of the bulk analysis significantly improves the paper, and shows the value of the entire work. One minor point that still needs improvement, is the description of the results in terms of protein numbers (per cell vs. TMT set vs. experiment). They indicated it properly in most cases, but there are still cases that this was not clear and should be stated, at least in the figure legend. However, this is only a technical comment.

Overall, I think the study in its current form shows better results than previous SCP studies, and also shows the biological value of such analyses.

REVIEWER COMMENTS

Reviewer #1 (Remarks to the Author):

I will here address the point-by-point discussion provided by the authors.

1. Thank you for the clarification.
2. Thank you for taking the time to test Fido and Dart-ID. It is interesting, and possibly not surprising, to see that proteins with only unique peptide hits are rescued.
3. The authors have tested the effect of imputing some values, which is useful given the large number of missing values in single cell MS-based experiments. They assess the impact of imputation on these data for different filtering criteria (Fig 5b), using silhouette scores, which I find a very useful approach. I was however initially surprised not to see any silhouette plots in the data after the repeated mention of silhouettes, and it took me some time to understand that they used the mean silhouette score. Typically, scores are computed for each data point in the set to be clustered, and all these are then visualised on a plot. Scores can also be averaged for each cluster to provide cluster-level validation metrics. These metrics and visualisations could certainly be made use of. The authors should at least clarify in the main text that they use the mean value (it does appear in the material and methods later), and ideally acknowledge that they then miss the variation within that score. See also below.

We thank the reviewer for their great suggestion to show all Silhouette Coefficients instead of the mean. We now show Boxplots of the distribution of the Silhouette Coefficients in Figure 2e, 3b and 5d. We also adapted the color scheme in Figure 2d. For Supplementary Figure 5b and the similar Supplementary Figure 7a, we decided to keep showing only the mean of all coefficients, as we find the boxplots to be suboptimal to judge the small differences in the coefficient distributions (See RebuttalFig1). Furthermore, we clarified the use of the mean score in the main text and method section where relevant.

RebuttalFig1 Comparison of the visualization of the silhouette coefficients with a mean point plot and boxplots.

4. It is unclear how 5 relates to the additional GO/pathway enrichment analyses. In Fig 4, I see that the box under the volcano plot 2b is indeed relevant.

The protein clusters shown in the heatmap in Figure 5c are annotated with enriched GO/pathway terms to the right of the plot.

5 and 6. Thank you for making a Python package available. It would still benefit from some user-level documentation. Scanpy and scanorama, which the authors refer to, are good examples thereof.

We thank the reviewer for their good suggestion and improved the documentation in the README file.

7. I have looked at the create_meta_data function. It looks like the function, and with it SCEPTRE, are focused on Proteome Discoverer output files (see also line 201). If so, please mention this. Otherwise, please document how create_meta_data (or load_dataset) can be used more generally.

SCEPTRE is indeed focused on Proteome Discoverer output files, which is mentioned on line 206. To help clarify this, we also document the use of both functions more in detail in the README file.

8. Overlay and annotate are spot on.

9. As discussed above, while using silhouette scores and plots is an useful approach, the implementation is a bit inelegant. I would suggest, at least, to visualise the distributions of the individual scores in Figure 3b, rather than the inadequate bar plots. The comparison of silhouette score distributions also applies to supplementary figure 5b.

We thank the reviewer for the suggestion, which we have implemented now (see answer to point #3).

10. Thank you for the clarification.

11. Thank you for the update. I also appreciate the usage of scanorama. I am not sure that the claims of "unbiased removal of batch effect" are totally appropriate (lines 97 and 224). You have demonstrated batch removal, but doing so without (some) bias is unlikely and more difficult to demonstrate.

We thank the referee for this point; we have rephrased to reflect the fact that we indeed likely cannot remove any bias entirely (lines 97, 229).

12. Thank you for the clarification and the rephrasing.

13. Thank you for your clarifications. I understand that your iterative procedure does remove unwanted technical/batch variances while keeping biological variance, but I have a hard time appreciating that it is a very good one. It would be interesting to compare it with other approaches, albeit this is probably out of the scope of this manuscript.

We agree with the reviewer that such evaluation would be of interest, and will explore other approaches for future work.

14. It is reassuring that both approaches return comparable results.

15. Thank you for the clarification.

16. I am not sure whether 5 nearest neighbours is any better than 2 for imputation. Ideally, the authors would apply a procedure as shown on their supplementary figure 5b with different values of k, thus formalising and validating their choice.

We thank the reviewer for their comments. We subsequently tested different numbers of nearest neighbors for the imputation, which is supported by the 'find_embedding_params' function in Sceptre. Here we found that knn=5 is indeed the better parameter choice compared to knn=3 and knn=10, resulting in no change of the selected fraction of valid values filter for the medium, high and bulk datasets. (See RebuttalFigure 2).

RebuttalFigure 2 Determination of the optimal fraction of valid values filter for the imputation for the embedding.

17 and 18. Thank you for the clarification. See my comments on silhouette scores/plots above.

20. As mentioned above, the README file should be completed with documentation of the package or at least a description/enumeration of its the functions. It could also mention the MIT license.

We appreciate the reviewer's suggestions, and, as also remarked in points 5, 6 & 7, we now improved the documentation in the README file and mention the MIT licence on line 718. On github, the MIT licence will be shown automatically in the About tab.

I got access to SCeptre through PRIDE; this is not an adequate way to distribute code. I assume this is a temporary means until the Github repository becomes available. Please remove the SCeptre.zip file before publication of the data and Github repository to avoid confusion; users are likely to use that static version and miss new features and bug fixes.

We thank the reviewer for their request of additional clarification. Indeed sharing the repository via PRIDE as the SCeptre.zip was a temporary workaround during the manuscript submission, as we the repository is now active on Github. Only the required data is placed in the PRIDE repository due to its file size prohibiting inclusion on github.

Other comments:

- Please update reference 25. SCoPE2 has now been published (<https://doi.org/10.1186/s13059-021-02267-5>).

- This is probably my limited expertise in biochemistry speaking for itself here, but it would be useful to define TFE which. Unless I missed it, it isn't defined.

- Supplementary tables 1 and 3 should be proper tables, not as pdf files.

Laurent Gatto

We thank the reviewer for these final comments and have updated the manuscript where relevant.

Reviewer #2 (Remarks to the Author):

The authors have produced a much improved manuscript – although I note it has now switched journals.

I am pleased that the authors have answered many of my concerns in their revised manuscript. One set of revisions I feel are still very misleading and this surrounds

quantitative accuracy.

To address the issue of accuracy I think the authors carried out an experiment where they compared blasts versus LSC cells at the single cell level and also blasts versus LSCs using bulk cells in a typical TMT based proteomics workflow. Both experiments gave fold changes in abundance between the two sets of cells and the authors compared whether fold changes acquired in the single cell experiment with what was achieved in bulk cell experiment using Pearson correlation. The experimental design is poorly described in the text and I have no idea how the data presented in the figures listed below are related to one another. The explanation given on lines 181-183 was not clear to me.

The results of the comparisons, as far as I understand are presented as the following:

Figure 3 - Pearson correlation of the protein fold changes but only for the top 400 high-coverage proteins

Figure S3C – Pearson Correlation between fold changes for all injection times, but only proteins that have no missing values (top) or that are present in all experiment (bottom)

Figure S5C – Pearson correlation now just for 300ms and 500ms where non-imputed values were used and only proteins with $n \geq 3$ in blasts and LSCs respectively were considered

These data were also presented in figure 2d in a different form.

In the rebuttal letter the authors also note that there is a lower dynamic range of fold changes observed in single cell data than the bulk data and note improved accuracy – as judged by Pearson correlation scores – when longer inject times are used.

I think the data as presented are very misleading. Pearson correlation scores are not a substitute for accuracy. In simple terms, if a fold change of 2 is measured for a pair of proteins in the bulk data, is the fold change of the same pair of proteins 2 measured in the single cell data comparable? Showing a linear correlation does not show that the same fold change values are achievable by both the bulk and single cell workflows. The Pearson correlation figures show a lot of scatter of data off the diagonal and no measure of the variability of the measurements across the intensity range of the measurements. I think the technical variability of the system is not fully explored here. Moreover, figure 3C in the main text portrays basically cherry picked data – i.e. the top 400 most intense measurements. Figure S5C shows a very different picture but is hidden in the supplementary data. I suggest that FigS5C should be promoted into the main body of the manuscript along with a fuller explanation of accuracy issues in the text.

The authors need to do a much better job depicting the accuracy of the single cell data. Additionally, in the main text they need to note the differences in dynamic range of fold change measurements and look for systematic bias in fold change measurements across the intensity range.

We thank the reviewer for their fair criticisms and request for further clarifications in terms of the accuracy of our data. Their interpretation of our initial attempt was fully accurate, and we hope that with this second attempt, we have managed to address this aspect more accurately and appreciate the good suggestions.

We added a clarification of the experimental design of the MS3 bulk-sorted data to the main text to ensure this was explained in more detail (lines 183-188). We would also like to clarify that the analysis in Figure 2d and Supplementary Figure 3 is

performed on the pooled 'single-cell' samples (lines 188-195), rather than the "real" single cell samples described in Figure 3 and S5. The analysis in Figure 3 and Supplementary Figure 5 is performed on our real scMS datasets with the 'medium' and 'high' method (lines 267-273).

We fully agree with the reviewer that the Pearson correlation alone is not sufficient to evaluate quantitative accuracy. Therefore, we adapted Figure 3 and added additional analyses. As kindly suggested by the reviewer, we now show the Pearson correlation of all proteins for the 'medium' and 'high' dataset in panel c (previously in FigS5C). Additionally, in order to clarify that high abundant proteins tend to be more accurately quantified, we still show the selection of the top 400 proteins for each dataset in panel d. In panel e, we investigate more in depth how the absolute difference between protein fold changes in scMS data and MS3 bulk-sorted data varies across the intensity range. Here we discovered that while accuracy decreases with lower intensity, the 'high' method consistently outperforms the 'medium' method across all intensity bins. Finally, in panel f we clarify that fold changes in scMS tend to be lower than the fold changes measured in MS3 bulk-sorted data. We do this by showing selected proteins across the scMS intensity range that were differentially expressed with high absolute fold changes in the MS3 bulk-sorted data. It becomes apparent that scMS is limited in its dynamic range, however the overall directions of the fold changes are in agreement and furthermore, the 'high' dataset shows a higher dynamic range than the 'medium' dataset.

For the analysis of the pooled 'single-cell' samples in Figure 2 we think that the Pearson correlation is a good measure to compare the different ion injection times for the initial determination of an appropriate MS instrument parameter range. We therefore decided to keep the data in Figure 2 as is and hope that the reviewer agrees that the in-depth analysis with real scMS datasets in Figure 3 now better clarifies the accuracy issues surrounding scMS data (lines 267-273).

Thorough analysis of

A) Number of cells in the booster channel

B) FAIMS settings

Have been discounted as outside the scope of the manuscript and the authors seem happy to go with values suggested by other labs. In the case of the latter I agree with their sentiments. For the former however, every cell line may be different in terms of the amounts of protein recovered in the workflow, and 200 cells as a booster channel may be OK for one cell types but not for the next. This needs further exploration in the text.

We thank the referee for this additional request of clarification, and have now addressed the requirement of careful evaluation of booster channel amount in every experimental setting individually in the text. We agree that depending on cell size, complexity of the cell mixture, etc., different booster amounts might be appropriate and needs to be very carefully controlled. More concretely, in lines 437-439 we now specify that:

In general, the amount of boosting should be carefully evaluated in each experimental setting in terms of its impact on the single-cell quantitative accuracy^{35,36}.

Reviewer #3 (Remarks to the Author):

In their revised manuscript the authors substantially improved the study, and I now find it suitable for publication in Nature Communications.

In my view the addition of the bulk analysis significantly improves the paper, and shows the value of the entire work. One minor point that still needs improvement, is the description of the results in terms of protein numbers (per cell vs. TMT set vs. experiment). They indicated it properly in most cases, but there are still cases that this was not clear and should be stated, at least in the figure legend.

We thank the reviewer for their careful evaluation of our manuscript, and have attempted to update the figure legends where there could still be doubt.

However, this is only a technical comment.

Overall, I think the study in its current form shows better results than previous SCP studies, and also shows the biological value of such analyses.

We are very appreciative of these kind words and are glad that the reviewer agrees with the potential of scMS for biology.

REVIEWERS' COMMENTS

Reviewer #2 (Remarks to the Author):

The authors have done an excellent job answering my queries and providing additional analysis of their data.

I have no other comments and consider the manuscript suitable for publication.

I agree with reviewer #3 that this is a study of high quality compared with some of the other single cell proteomics manuscripts in the public domain

The authors should be congratulated for such a thorough study